# Bladder Cancer and Risk Factors: Data from a Multi-Institutional Long-Term Analysis on Cardiovascular Disease and Cancer Incidence

**DOI:** 10.3390/jpm13030512

**Published:** 2023-03-13

**Authors:** Biagio Barone, Marco Finati, Francesco Cinelli, Antonio Fanelli, Francesco Del Giudice, Ettore De Berardinis, Alessandro Sciarra, Gianluca Russo, Vito Mancini, Nicola D’Altilia, Matteo Ferro, Angelo Porreca, Benjamin I. Chung, Satvir Basran, Carlo Bettocchi, Luigi Cormio, Ciro Imbimbo, Giuseppe Carrieri, Felice Crocetto, Gian Maria Busetto

**Affiliations:** 1Department of Neurosciences, Reproductive Sciences and Odontostomatology, University of Naples “Federico II”, 80131 Naples, Italy; 2Department of Urology and Renal Transplantation, University of Foggia, 71122 Foggia, Italy; 3Department of Maternal-Infant and Urological Sciences, Sapienza Rome University, 00185 Rome, Italy; 4Department of Public Health, University of Naples “Federico II”, 80131 Naples, Italy; 5Department of Urology, European Institute of Oncology (IEO) IRCCS, 20141 Milan, Italy; 6Oncological Urology, Veneto Institute of Oncology (IOV) IRCCS, 35128 Padua, Italy; 7Department of Urology, Stanford Medical Center, Stanford, CA 94305, USA

**Keywords:** bladder cancer, cardiovascular disease, risk factors, comorbidities

## Abstract

**Background:** Bladder cancer (BCa) is a heterogeneous disease with a variable prognosis and natural history. Cardiovascular disease (CVD), although completely different, has several similarities and possible interactions with cancer. The association between them is still unknown, but common risk factors between the two suggest a shared biology. **Materials and Methods:** This was a retrospective study that included patients who underwent transurethral resection of bladder tumor at two high-volume institutions. Depending on the presence of a previous history of CVD or not, patients were divided into two groups. **Results:** A total of 2050 patients were included, and 1638 (81.3%) were diagnosed with bladder cancer. Regarding comorbidities, the most common were hypertension (59.9%), cardiovascular disease (23.4%) and diabetes (22.4%). At univariate analysis, independent risk factors for bladder cancer were age and male sex, while protective factors were cessation of smoking and presence of CVD. All these results, except for ex-smoker status, were confirmed at the multivariate analysis. Another analysis was performed for patients with high-risk bladder cancer and, in this case, the role of CVD was not statistically significant. **Conclusions:** Our study pointed out a positive association between CVD and BCa incidence; CVD was an independent protective factor for BCa. This effect was not confirmed for high-risk tumors. Several biological and genomics mechanisms clearly contribute to the onset of both diseases, suggesting a possible shared disease pathway and highlighting the complex interplay of cancer and CVD. CVD treatment can involve different drugs with a possible effect on cancer incidence, but, to date, findings are still inconclusive.

## 1. Introduction

Bladder cancer (BCa) is a heterogeneous disease with a variable prognosis and natural history [1]. It comprises non-muscle invasive disease (NMIBC), i.e., bladder cancers that do not involve the muscular layer of the bladder, and muscle-invasive disease (MIBC) which instead is characterized by the invasion of the muscle layer of the bladder. More than 70% of bladder cancers are diagnosed as NMIBCs. For MIBC, it is important to identify those at higher risk to better control future disease recurrence and progression [2,3,4]. In a recent meta-analysis, tobacco smoking and occupational exposure remain the most important risk factors for bladder carcinogenesis [5,6], but unfortunately, no protective factors have yet been identified.

Cardiovascular disease (CVD) is a leading cause of death worldwide, and although seemingly unrelated, CVD and cancer have several similarities and possible interactions [7]. The association between cardiovascular disease and cancer is still unknown, but risk factors, such as hypertension, obesity and diabetes, can be common, suggesting a shared biology, currently under evaluation [8]. Inflammation is an important mechanism as it is reported to be a possible trigger for both carcinogenesis and CVD. If we consider, once again, that common risk factors such as obesity, hyperglycemia, hypertension, and hypertriglyceridemia induce inflammation, then this could connect cancer and CVD [9,10]. Vincent et al., in a recent review, highlighted shared disease pathways which overlap in risk factors between cancer and CVD. The authors offered a framework for a system-based approach to reduce overall risk burden, providing opportunities for joint risk-factor modification [10].

Another potential etiologic factor is medications and, in our aging population, medications prescribed for cardiovascular indications are becoming more and more common. Anti-hypertensives, anticoagulants, and statins, used alone or in combination, are potential factors. However, their action, particularly on bladder cancer carcinogenesis, is still not fully understood [11].

There are no conclusive correlations between cardiovascular disease and bladder cancer in the literature and no data are reported to analyze if BCa incidence is increased or decreased in patients suffering from CVD. Furthermore, no relationship between high- or low-risk BCa and CVD has been reported. The only related evidence has been reported by Kok et al. who performed a population-based cohort study on 39,618 adults. They observed that patients suffering from hypertension had a 32% increased risk of developing bladder cancer, but when examined by gender, this result was statistically significant only for females [12]. Therefore, our aim was to further evaluate the relationship between bladder cancer and cardiovascular disease in a multi-institutional study, including more than 2000 BCa patients.

## 2. Materials and Methods

This was a retrospective study approved by the institutional review board of “Policlinico Riuniti of Foggia” with protocol # 31/CE/2022 (DCS #7) on 28th of February 2022 and conducted in accordance with the World Medical Association Declaration of Helsinki. All patients included provided written informed consent for the procedures as well as for the participation and publication of the study. Patients involved underwent transurethral resection of bladder tumors (TURBT) at two large volume institutions—University of Naples “Federico II” and University of Foggia—between 1st of January 2008 and 31st of December 2021. Eligibility criteria were age >18 years and a previous ultrasound, cytology, or cystoscopy suggested a potential malignancy. Every patient underwent a cystoscopy prior to TURBT, in accordance with European Association of Urology (EAU) guidelines. All clinical and laboratory data, as well as patient’s history, such as comorbidities and therapies, were retrieved and analyzed from medical records. All surgical specimens were processed, described, and reviewed by dedicated uropathologists. For urothelial cancer, grade was classified according to the WHO/ISUP 2016 grading system [13]. Pathological stage was assigned following the current American Joint Committee on Cancer 2017 TNM staging system (VIII edition) [14]. Patients who exhibited high-grade (HG) NMIBC or MIBC underwent staging with abdominal-pelvis computed tomography (CT) scan with contrast, chest CT scan or X-ray, and bone scan. Second-level examinations, such as magnetic resonance imaging (MRI) or total body positron emission tomography (PET)-CT were performed only in case of clinical suspicion or symptoms. For a previous history of CVD, patients were divided into two groups: Group A, which included all the patients who had a history of confirmed CVD (such as stroke, heart failure, valvular heart disease) or had undergone related surgery (such as percutaneous transluminal coronary angioplasty); Group B included all the patients who did not report similar conditions. Another sub-classification was carried out, dividing high-risk tumors from the others. The majority of patients involved were of Caucasian ethnicity. All data are reported in Table 1.

### Statistical Analysis

Descriptive statistics were reported as means and standard deviations for continuous variables while frequency and percentages were reported for categorical variables. According to the normality of data, assessed via the Kolmogorov–Smirnov test, T-test and Mann–Whitney U test were used for group comparisons for continuous variables. Similarly, categorical variables were compared between the two groups via the Chi-square test. Finally, univariate and multivariate logistic regression were used in order to obtain the odds ratio (OR), and the corresponding 95% confidence interval (CI), for protective/risk factors for any bladder cancer and HG NMIBC. All statistical analyses were performed using IBM SPSS software (version 25, IBM Corp, Armonk, NY, USA) and considering *p* < 0.05 as statistically significant.

## 3. Results

A total of 2050 patients were included in the study. The descriptive statistics of the patients involved in the study are reported in Table 1. Of the patients, 577 (28.1%) were non-smokers, 865 (42.2%) ex-smokers, and 578 (28.2%) active smokers. Regarding other comorbidities, the most common were hypertension (59.9%), followed by cardiovascular disease (*n* = 479, 23.4%), and diabetes (22.4%). A total of 1638 (81.3%) patients reported bladder cancer. In particular, 333 (16.2%) had grade 1 disease, 425 (20.7%) grade 2, and 880 (42.9%) grade 3. Multifocality was reported in 544 (26.5%) of cancers, while concomitant carcinoma in situ was reported in 64 (3.1%). Median follow-up was 2.11 years.

When patients were compared regarding their CVD status, both groups were comparable in terms of years of smoking, years without smoking, cigarettes per day, and a history of previous cancers. Conversely, patients with CVD (Group A: 479 patients) were older than those without CVD (Group B: 1569 pts) with a mean of 73.88 ± 8.87 vs. 69.79 ± 12.11 (*p* < 0.0001), as well as a slightly higher prevalence among males (89.8% vs. 75.1%, *p* < 0.0001). Group A had a higher prevalence of comorbidities compared to Group B, and reported a higher rate of ex-smokers. Interestingly, the CVD group reported a lower rate of bladder cancer with 368 (78.3%) vs. 1269 (82.2%) in Group B (*p* = 0.048), despite a higher presence of grade 3 cancer (46.4% vs. 42.8%, *p* = 0.024) and slightly higher pathological stages, with 119 (26.4%) vs. 306 (20.1%) for pT1, 56 (12.4%) vs. 186 (12.2%) for pT2 and 2 (0.4%9 vs. 2 (0.1%) for pT4 (*p* = 0.005). Data are reported in Table 2, Figure 1 and Figure 2. 

According to these results, in order to clarify the role of other statistically significant comorbidities in the two groups as potential risk factors for bladder cancer, we performed a univariate and multivariate logistic regression, with regard to the presence or absence of any bladder cancer, in the entire cohort. As reported in Table 3, independent risk factors for bladder cancer were age (OR = 1.037, 95% CI 1.027–1.047, *p* < 0.0001) and male sex (OR = 1.675, 95% CI 1.263–2.222, *p* < 0.0001) while protective factors were the cessation of smoking (OR = 0.743, 95% CI 0.569–0.970, *p* = 0.029) and the presence of CVD (OR = 0.781, 95% CI 0.605–1.008, *p* = 0.047). All these results, except for ex-smoker status, were confirmed with multivariate analysis, reporting, in particular, an even lower odds ratio for CVD (OR = 0.659, 95% CI 0.496–0.875, *p* = 0.004).

At this point, to further clarify the protective role of CVD in BCa, we performed a univariate and multivariate logistic regression for both groups (Table 4). Apart from age and male sex which reasonably influenced the odds ratio for BCa, only COPD, in the group of patients with CVD, was statistically significant among the other evaluated risk factors, reporting OR = 2.447 (95% CI 1.250–4.788, *p* < 0.0001) and OR = 2.863 (95% CI, 1.359–6.028, *p* = 0.006) on univariate and multivariate analysis, respectively.

A similar analysis was performed for patients with high-risk BCa. As reported in Table 5, in this case, the role of CVD was not statistically significant, while age and male sex continued to be independent predictors of HG BCa on multivariate analysis with an OR = 1.056 (95% CI 1.040–1.072, *p* < 0.0001) and OR = 2.088 (95% CI 1.343–3.248, *p* = 0.001). Similar results were reported for the univariate and multivariate logistic regression analyses stratified for the two groups (Table 6); among patients with CVD (Group A), age and COPD were associated with high-risk BCa, reporting an OR = 1.047 (95% CI 1.009–1.087, *p* = 0.016) and OR = 2.993 (95% CI 1.264–7.086, *p* = 0.013), respectively; for patients without CVD, the only predictors for HG BCa were, similarly, age and male sex, reporting in the multivariate analysis an OR = 1.058 (95% CI 1.040–1.076, *p* < 0.0001) and OR = 2.029 (95% CI 1.239–3.322, *p* = 0.005), respectively.

## 4. Discussion

This multi-institutional retrospective analysis was conducted on a large population of more than 2000 patients that underwent trans-urethral resection of the bladder and all procedures were performed at two academic high-volume urologic oncologic centers. In our study, more than 81% were diagnosed with bladder cancer, confirmed at pathological evaluation. Our study pointed out a positive association between CVD and BCa incidence; univariate and multivariate analysis confirmed that CVD was an independent protective factor for BCa. This effect, reported for the entire population, was not confirmed for the high-risk tumors. Patients with CVD also exhibited a higher prevalence of other comorbidities, such as diabetes and COPD, as well as a history of smoking. Nevertheless, the presence of CVD independently predicted the incidence of BCa, but not its aggressiveness.

Another interesting point was the differential impact of comorbidities among BCa patients, depending on tumor staging. Specifically, a reduced risk of harboring BCa was observed in patients with CVD, but this statement was not valid for high-risk tumors. The evolution of BCa genomics could possibly explain the relationship between different tumor histotypes and risk factors. BCa is characterized by a high mutational heterogeneity, with frequent mutations in multiple different signaling pathways, including cell-cycle genes, tyrosine kinase receptors, PI3K/AKT/mTOR, and chromatin regulatory gene mutations [15]. Some of these pathways, such as fibroblast growth factor signaling, contribute also to atherosclerosis by enhancing an inflammatory response in vascular smooth muscle [16]. FGFR3 mutations are more common in non-invasive low grade papillary tumors and could possibly explain the different impact of CVD on this heterogeneous and complex disease [17]. It is well known that low-risk cancers and high-risk cancers follow two completely different pathways. On one side, altered cells follow hyperplasia that evolve toward low-grade tumors, on the other, they become dysplastic (tp53 mutation) and follow the CIS pathway that could evolve toward invasive carcinoma [18]. Probably, even this mechanism should be taken into consideration to better understand why CVD is not able to perform its protective effect over high-risk tumors.

A possible link between CVD and BCa has been proposed in the past few years, although no conclusive evidence has been established yet [7,19,20,21]. A significant overlap in epidemiology and modifiable risk factors would prove how both the screening and management of certain chronic conditions can reduce the downstream risk of both cardiovascular disease and incident BCa [9,22]. Further studies are required to evaluate, in addition to the role of CVD, the influence of other variables as age, gender and body mass index [23].

Moreover, several biological and genomics mechanisms clearly contribute to the onset of both diseases, suggesting a possible shared disease pathway and highlighting the complex interplay of cancer and CVD [16]. Chronic inflammation and oxidative stress, for example, could promote the expression of the adhesion molecules and growth factors necessary to not only foster atherosclerosis, but also malignancies [24]. The inflammatory intersection is further reinforced by recent findings in the CANTOS trial (canakinumab anti-inflammatory thrombosis outcomes study), where the specific targeting of interleukin 1β (a key mediator of inflammation) conferred cardiovascular benefit and, unexpectedly, a significant decrease in cancer incidence [25].

CVD treatment and control generally involve a multitude of drugs and related drug-interactions, with a possible effect on cancer incidence. Considering BCa risk, several studies have considered the relationship with aspirin but have demonstrated inconsistent results and overall null associations [26,27,28]. Similar results were assessed for any antihypertensive agents or statin use in a population-based case-control study, highlighting the need for additional long-term follow-up prospective research [18,29,30]. A possible protective role related to certain CVD medications could explain the differential impact in our cohort. A second-level analysis evaluating the association of BCa and drug prescription in our multicenter series will be the object of further studies.

To the best of our knowledge, this paper is the first to highlight a possible link between CVD and BCa carcinogenesis and differentiation. Less than 20% of our population who underwent TURBT in our multicenter series did not exhibit BCa, while chronic inflammation was the most common finding at pathologic examination. Patients with CVD are usually at higher risk of exhibiting hematuria due to antiplatelet/antithrombotic drugs, which often requires second-level examination such as ultrasound, cystoscopy or a computed tomography scan [31]. Performing such diagnostic procedures could also increase the detection of benign pseudocarcinomatous lesions. Bladder epithelial proliferations such as chronic cystitis, are most typically seen in association with radiation or chemotherapy, but a recent study found that in almost 90% of cases there was still a possible etiology for proliferation in the form of localized ischemia and peripheral vascular disease [32]. This relationship could possibly justify the lower incidence of BCa in the CVD group, although the higher incidence of high-risk tumors seems to underline a different oncologic pattern between the two groups. Further prospective studies are required to truly assess the impact of CVD on bladder carcinogenesis.

The major strength of our study is the large population enrolled and the novelty of our findings. Several limitations should be highlighted: retrospective data, absence of an analysis that correlates the results with patient medication, and the inability to include the general population as a control group. 

## Figures and Tables

**Figure 1 jpm-13-00512-f001:**
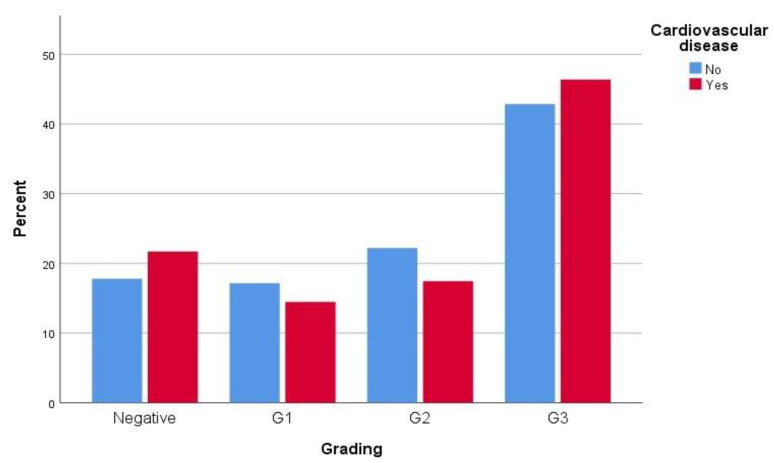
Distribution of BCa grading according to presence of CVD.

**Figure 2 jpm-13-00512-f002:**
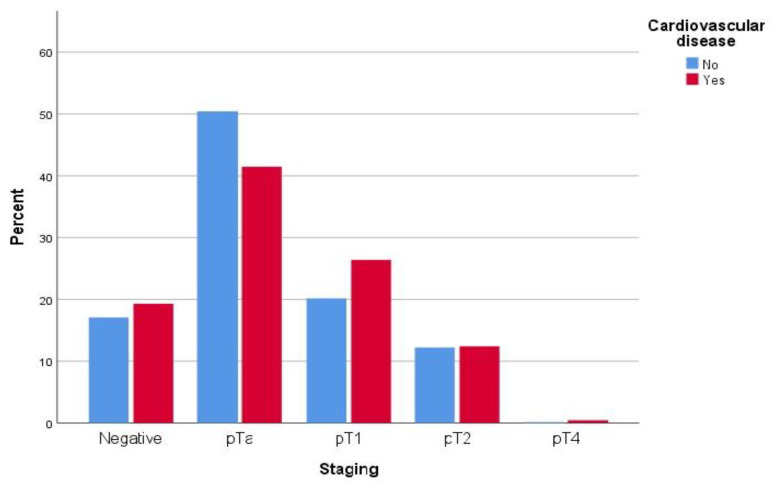
BCa tumor staging distribution according to presence of CVD.

**Table 1 jpm-13-00512-t001:** Descriptive characteristics of patients involved.

Variable	Mean	Standard Deviation
**Age**	70.73	11.56
**Years of smoking**	42.57	12.80
**Years without smoking**	16.64	12.35
**Number of previous TURBT**	0.37	1.065
**Number of total TURBT**	2.2	1.67
	**Count**	**Percentage**
**Sex**		
Male	1730	84.4
Female	320	15.6
**Smoker**		
Yes	578	28.2
No	577	28.1
Ex	865	42.2
**Diabetes**	460	22.4
**Hypertension**	1227	59.9
**Dyslipidemia**	431	21
**COPD**	273	13.3
**Cardiovascular disease**	479	23.4
**Chronic kidney disease**	143	7
**Hepatitis C virus**	53	2.6
**Other cancers**	326	15.9
**Prostate cancer**	77	3.8
**Kidney cancer**	31	1.5
**Upper tract urinary cancer**	34	1.7
**Recurrent tumor**	130	6.3
**Bladder cancer**	1638	81.3
**Grading**		
G1	333	16.2
G2	425	20.7
G3	880	42.9
**Staging**		
pTa	955	46.6
pT1	426	20.8
pT2	242	11.8
pT4	4	0.2
**Multifocal**	544	26.5
**Concomitant CIS**	64	3.1
**MIBC**	242	11.8
**Treatment**		
MMC	152	7.4
BCG	255	12.4
Cystectomy	50	2.4
Re-TURBT	115	5.6

**Table 2 jpm-13-00512-t002:** Differences between patients with (Group A) and without (Group B) cardiovascular disease (CVD).

	Group A (*n* = 480)	Group B (*n* = 1570)	*p* Value
**Age**	73.88 ± 8.87	69.79 ± 12.11	**<0.0001**
**Years of smoking**	44.03 ± 10.96	41.98 ± 13.45	0.241
**Years without smoking**	17.08 ± 11.62	16.34 ± 12.83	0.608
**Cigarettes per diem**	21.11 ± 15.27	20.23 ± 12.57	0.478
**Number of previous TURBT**	0.53 ± 1.17	0.32 ± 1.02	**<0.0001**
**Number of total TURBT**	2.27 ± 1.64	2.18 ± 1.68	0.310
**Sex**			
Male	431 (89.8)	1299 (82.7)	**<0.0001**
Female	49 (10.2)	271 (17.3)	**<0.0001**
**Smoker**			
Yes	120 (25.7)	456 (29.4)	**<0.0001**
No	100 (21.4)	477 (30.8)	**<0.0001**
Ex	247 (52.9)	618 (39.8)	**<0.0001**
**Diabetes**	157 (32.8)	302 (19.2)	**<0.0001**
**Hypertension**	353 (73.7)	873 (55.6)	**<0.0001**
**Dyslipidemia**	148 (34.4)	282 (18)	**<0.0001**
**COPD**	99 (20.7)	173 (11)	**<0.0001**
**Chronic kidney disease**	51 (10.7)	92 (5.9)	**<0.0001**
**Hepatitis C virus**	17 (3.6)	36 (2.3)	0.126
**Other cancers**	72 (18.8)	254 (22.6)	0.117
**Prostate cancer**	15 (4.5)	62 (7.8)	0.47
**Kidney cancer**	11 (3.3)	20 (2.5)	0.462
**Upper tract urinary cancer**	7 (2.1)	27 (3.4)	0.245
**Recurrent tumor**	21 (9.3)	109 (10.4)	0.622
**Bladder cancer**	368 (76.7)	1270 (80.9)	0.048
**Grading**			
G1	68 (14.5)	265 (17.2)	**0.024**
G2	82 (17.4)	343 (22.2)	**0.024**
G3	218 (46.4)	661 (42.8)	**0.024**
**Staging**			
pTa	187 (41.5)	768 (50.5)	**0.005**
pT1	119 (26.4)	306 (20.1)	**0.005**
pT2	56 (12.4)	186 (12.2)	**0.005**
pT4	2 (0.4)	2 (0.1)	**0.005**
**Multifocal**	144 (34.1)	400 (29.7)	0.085
**Concomitant CIS**	22 (5.4)	42 (3.5)	0.087
**MIBC**	58 (12.5)	188 (12.3)	0.438
**Treatment**			
MMC	39 (8.4)	113 (7.7)	0.764
BCG	68 (14.4)	187 (12.7)	0.764
Cystectomy	11 (2.4)	39 (2.6)	0.764
Re-TURBT	28 (6.1)	87 (5.9)	0.764

**Table 3 jpm-13-00512-t003:** Univariate and multivariate logistic regression analysis for bladder cancer (BCa) risk factor in the entire cohort.

Variable	Univariate (95% CI)	*p* Value	Multivariate	*p* Value
**Age**	1.037 (1.027–1.047)	**<0.0001**	1.040 (1.029–1.051)	**<0.0001**
**Male sex**	1.675 (1.263–2.222)	**<0.0001**	1.691 (1.242–2.303)	**0.001**
**Ex-smoker**	0.743 (0.569–0.970)	**0.029**	0.885 (0.653–1.201)	0.433
**Diabetes**	1.106 (0.842–1.452)	0.468	1.008 (0.750–1.353)	0.959
**Hypertension**	1.052 (0.838–1.321)	0.662	0.22 (0.635–1.064)	0.136
**Dyslipidemia**	1.189 (0.894–1.582)	0.234	1.226 (0.901–1.667)	0.194
**COPD**	1.292 (0.907–1.841)	0.155	1.203 (0.825–1.754)	0.336
**Chronic kidney disease**	0.763 (0.505–1.151)	0.763	0.666 (0.434–1.024)	0.64
**Cardiovascular disease**	0.781 (0.605–1.008)	**0.047**	0.659 (0.496–0.875)	**0.004**

**Table 4 jpm-13-00512-t004:** Univariate and multivariate logistic regression analysis for BCa risk factors in patients with and without CVD.

**Group A**				
**Variable**	**Univariate (95% CI)**	***p* Value**	**Multivariate**	***p* Value**
**Age**	1.035 (1.010–1.062)	**0.006**	1.027 (1.000–1.055)	0.53
**Male sex**	2.096 (1.093–4.020)	**0.026**	2.089 (0.972–4.489)	0.59
**Ex-smoker**	0.675 (0.360–1.268)	0.222	0.593 (0.302–1.164)	0.150
**Diabetes**	1.024 (0.641–1.636)	0.920	0.861 (0.520–1.425)	0.560
**Hypertension**	1.158 (0.709–1.890)	0.557	0.891 (0.509–1.559)	0.686
**Dyslipidemia**	1.295 (0.792–2.117)	0.303	1.326 (0.766–2.295)	0.313
**COPD**	2.447 (1.250–4.788)	**<0.0001**	2.863 (1.359–6.028)	**0.006**
**Chronic kidney disease**	0.961 (0.473–1.956)	0.913	0.793 (0.376–1.675)	0.544
**Group B**				
**Variable**	**Univariate (95% CI)**	***p* Value**	**Multivariate**	***p* Value**
**Age**	1.041 (1.030–1.052)	**<0.0001**	1.043 (1.031–1.055)	**<0.0001**
**Male sex**	1.656 (1.207–2.273)	**0.002**	1.613 (1.149–2.264)	**0.006**
**Ex-smoker**	1.084 (0.782–1.504)	0.628	0.991 (0.700–1.405)	0.961
**Diabetes**	1.219 (0.865–1.717)	0.258	1.086 (0.751–1.571)	0.660
**Hypertension**	1.074 (0.827–1.396)	0.593	0.797 (0.594–1.070)	0.131
**Dyslipidemia**	1.208 (0.847–1.723)	0.297	1.191 (0.817–1.735)	0.363
**COPD**	1.005 (0.661–1.526)	0.983	0.813 (0.524–1.262)	0.356
**Chronic kidney disease**	0.706 (0.425–1.171)	0.177	0.605 (0.357–1.026)	0.062

**Table 5 jpm-13-00512-t005:** Univariate and multivariate logistic regression analysis for high grade (HG) BCa risk factor in the entire cohort.

Variable	Univariate (95% CI)	*p* Value	Multivariate	*p* Value
**Age**	1.059 (1.044–1.074)	**<0.0001**	1.056 (1.040–1.072)	**<0.0001**
**Male sex**	1.996 (1.353–2.946)	**<0.0001**	2.088 (1.343–3.248)	**0.001**
**Ex-smoker**	0.572 (0.404–0.812)	**0.002**	0.759 (0.519–1.109)	0.759
**Diabetes**	1.038 (0.737–1.463)	0.830	0.915 (0.621–1.348)	0.915
**Hypertension**	1.403 (1.046–1.882)	**0.024**	1.016 (0.723–1.428)	0.926
**Dyslipidemia**	1.384 (0.978–1.959)	0.067	1.276 (0.861–1.893)	0.224
**COPD**	1.459 (0.954–2.232)	0.081	1.344 (0.844–2.140)	0.212
**Chronic kidney disease**	1.081 (0.656–1.780)	0.760	0.861 (0.502–1.475)	0.586
**Cardiovascular disease**	1.017 (0.741–1.395)	0.918	0.767 (0.536–1.097)	0.146

**Table 6 jpm-13-00512-t006:** Univariate and multivariate logistic regression analysis for HG BCa risk factors in patients with and without CVD.

**Group A**				
**Variable**	**Univariate (95% CI)**	***p* Value**	**Multivariate**	***p* Value**
**Age**	1.055 (1.020–1.091)	**0.002**	1.047 (1.009–1.087)	0.016
**Male sex**	1.842 (0.794–4.272)	0.155	2.259 (0.806–6.327)	0.121
**Ex-smoker**	0.963 (0.497–1.866)	0.911	1.070 (0.521–2.198)	0.854
**Diabetes**	0.948 (0.531–1.693)	0.812	0.924 (0.483–1.767)	0.812
**Hypertension**	1.571 (0.832–2.966)	0.164	1.037 (0.492–2.185)	0.923
**Dyslipidemia**	1.427 (0.790–2.581)	0.313	1.428 (0.714–2.854)	0.313
**COPD**	2.591 (1.206–5.570)	**0.015**	2.993 (1.264–7.086)	**0.013**
**Chronic kidney disease**	1.045 (0.439–2.487)	0.921	0.685 (0.262–1.794)	0.442
**Group B**				
**Variable**	**Univariate (95% CI)**	***p* Value**	**Multivariate**	***p* Value**
**Age**	1.062 (1.045–1.080)	**<0.0001**	1.058 (1.040–1.076)	**<0.0001**
**Male sex**	2.046 (1.317–3.178)	**0.001**	2.029 (1.239–3.322)	**0.005**
**Ex-smoker**	0.466 (0.308–0.705)	**<0.0001**	1.056 (0.676–1.650)	0.083
**Diabetes**	1.089 (0.706–1.680)	0.699	0.917 (0.562–1.497)	0.729
**Hypertension**	1.370 (0.978–1.918)	0.067	0.997 (0.678–1.467)	0.989
**Dyslipidemia**	1.367 (0.885–2.111)	0.159	1.229 (0.752–2.008)	0.410
**COPD**	1.093 (0.649–1.841)	0.983	0.909 (0.515–1.607)	0.743
**Chronic kidney disease**	1.099 (0.597–2.024)	0.761	0.941 (0.486–1.825)	0.858

## Data Availability

Data are available on request to the corresponding author.

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
