# Peer review of "Bladder Cancer and Risk Factors: Data from a Multi-Institutional Long-Term Analysis on Cardiovascular Disease and Cancer Incidence"

_jpm, 2023, doi:10.3390/jpm13030512_

Round 1

Reviewer 1 Report

Bladder Cancer and CVD relation is an actual concern for many scientists.

There are a few aspects that may be improved in your paper.

1. In the first part of the introduction section there are too many data that you can easily find in the first table. 

2. The discussion section should be more focused on your results.

Author Response

there is the file with reply

Reviewer 2 Report

In this retrospective study, Barone et al tried to find a correlation between the presence of cardiovascular disease and bladder cancer. 

Major comments:

-The authors should make sure that the numbers in the tables are accurate:

--According to Table 1, 2050 patients were included in the study. However, according to Table 2, Group A has 479 patients and Group B has 1569, a total of 2048. Why were the 2 patients excluded from the study?

--Table 2, male patients: 430 out of 479 of Group A : 430/479= 89.7% (correct) but then, for Group B: 1298/1569=82.7% and not 75.1%

--Table 2, bladder cancer: Group A: 368/479=76.8% and not 78.3%, Group B: 1269/1569=80.7% and not 82.2%

-Muscle invasive bladder cancer is the cancer that has grown into the muscle of the bladder, so stage T2 and above. According to Table 1, 242 patients were diagnosed with pT2, and 4 with pT4, a total of 242. However, 2 lines below that, it says that 225 patients had MIBC. How is MIBC defined in this study?

-How do the authors define "high risk BCa"? According to the first paragraph of page 8, the only included patients with T1G3, however there are other criteria for this group of patients, such as the presence of carcinoma in situ. Also the authors should make clear the distiction between NMIBC and MIBC wherever appropriate.

Minor comments:

- What is the race/ethnicity of the participants?

Author Response

there is the file with reply

Round 2

Reviewer 2 Report

I would like to thank the authors for addressing the comments. The manuscript is now suitable for publication.